# Bridging the Species Gap: Morphological and Molecular Comparison of Feline and Human Intestinal Carcinomas

**DOI:** 10.3390/cancers13235941

**Published:** 2021-11-25

**Authors:** Tanja Groll, Franziska Schopf, Daniela Denk, Carolin Mogler, Ulrike Schwittlick, Heike Aupperle-Lellbach, Sabrina Rim Jahan Sarker, Nicole Pfarr, Wilko Weichert, Kaspar Matiasek, Moritz Jesinghaus, Katja Steiger

**Affiliations:** 1Institute of Pathology, School of Medicine, Technical University of Munich (TUM), 81675 Munich, Germany; tanja.groll@tum.de (T.G.); f.schopf@campus.lmu.de (F.S.); daniela.denk@patho.vetmed.uni-muenchen.de (D.D.); carolin.mogler@tum.de (C.M.); sabrina.sarker@tum.de (S.R.J.S.); nicole.pfarr@tum.de (N.P.); wilko.weichert@tum.de (W.W.); moritz.jesinghaus@uni-marburg.de (M.J.); 2Comparative Experimental Pathology, School of Medicine, Technical University of Munich (TUM), 81675 Munich, Germany; 3Institute of Veterinary Pathology, Center for Clinical Veterinary Medicine, Ludwig-Maximilians-Universitaet (LMU), 80539 Munich, Germany; kaspar.matiasek@neuropathologie.de; 4LABOKLIN GmbH & Co. KG, 97688 Bad Kissingen, Germany; schwittlick@laboklin.com (U.S.); aupperle@laboklin.com (H.A.-L.); 5Department of Pathology, University Hospital Marburg, 35043 Marburg, Germany

**Keywords:** spontaneous feline intestinal tumors, comparative oncology, colorectal cancer, tumor budding, *CTNNB1*

## Abstract

**Simple Summary:**

Colorectal cancer (CRC) is the second leading cause of cancer deaths in humans (2020) but modeling late-stage human CRC, including high tumor budding and metastatic activity, experimentally in mouse models is a major challenge. In the present study, histopathological, immunohistochemical and molecular features of spontaneous intestinal carcinomas in cats were evaluated with a special focus on their potential applicability as a valuable model for human CRC. Feline intestinal tumors display aggressive growth patterns and adequately model invasive late-stage human CRC. They exhibit the same histological subtypes and display strikingly high tumor budding activity, both of which are highly significant prognostic factors in human CRC. Moreover, human and feline colorectal tumors harbor the same mutations of the *CTNNB1* gene, encoding β-catenin. Our data indicate that feline intestinal carcinomas constitute a valuable and promising in vivo model for human CRC. Further comparative oncological research, and especially investigation of the molecular landscape of feline intestinal neoplasms, is imperative.

**Abstract:**

Limited availability of in vivo experimental models for invasive colorectal cancer (CRC) including metastasis and high tumor budding activity is a major problem in colorectal cancer research. In order to compare feline and human intestinal carcinomas, tumors of 49 cats were histologically subtyped, graded and further characterized according to the human WHO classification. Subsequently, feline tumors were compared to a cohort of 1004 human CRC cases. Feline intestinal tumors closely resembled the human phenotype on a histomorphological level. In both species, adenocarcinoma not otherwise specified (ANOS) was the most common WHO subtype. In cats, the second most common subtype of the colon (36.4%), serrated adenocarcinoma (SAC), was overrepresented compared to human CRC (8.7%). Mucinous adenocarcinoma (MAC) was the second most common subtype of the small intestine (12.5%). Intriguingly, feline carcinomas, particularly small intestinal, were generally of high tumor budding (Bd) status (Bd3), which is designated an independent prognostic key factor in human CRC. We also investigated the relevance of feline *CTNNB1* exon 2 alterations by Sanger sequencing. In four cases of feline colonic malignancies (3 ANOS, 1 SAC), somatic missense mutations of feline *CTNNB1* (p.D32G, p.D32N, p.G34R, and p.S37F) were detected, indicating that mutational alterations of the WNT/β-catenin signaling pathway potentially play an essential role in feline intestinal tumorigenesis comparable to humans and dogs. These results indicate that spontaneous intestinal tumors of cats constitute a useful but so far underutilized model for human CRC. Our study provides a solid foundation for advanced comparative oncology studies and emphasizes the need for further (molecular) characterization of feline intestinal carcinomas.

## 1. Introduction

Worldwide, colon cancer is the third most common cancer type in humans, with an estimated 1.9 million new cases in 2020 and it is the second leading cause of cancer deaths [1]. To date, the most frequently used animal models in colon cancer research are mice with experimentally induced intestinal cancer; however, modeling late stages of human CRC is a major challenge as several mouse models tend to develop a high tumor burden, but no metastasis, leading to preliminary death. Invasively growing, metastasizing tumors of the large intestine, comparable with late-stage human CRC, are difficult to model in clean-housed experimental animal models, particularly genetically engineered mouse models (GEMM) [2,3,4,5]. Similarly, other animal models, e.g., rats or pigs, lack the ability to model metastasis and invasive carcinoma [3]. Moreover, the genotype of human tumors appears to be more heterogeneous than the one of experimentally induced murine tumors [6]. Spontaneously arising intestinal tumors of companion animals (pets) thus are of special interest for comparative research, especially since companion animals share their living environment with humans [7].

At present, there are only few studies of feline intestinal cancer, its biological behavior, clinical, histopathological, and molecular features. In cats, lymphoma is the most common intestinal neoplasm followed by intestinal carcinoma with a study-dependent incidence varying from 17 to 31.5% among gastrointestinal neoplasias [8,9,10]. Feline intestinal carcinomas are more prevalent than canine ones [11] and occur in both the large and small intestine. The available literature provides contradictory statements regarding the most commonly affected site in cats [8,9,11,12,13,14]. Feline intestinal carcinomas are more frequent in older animals with increasing risk starting from the age of seven [8,9] and either a breed predisposition of Siamese cats [9,11,13] or no breed predisposition [8,12] has been reported. Tumors metastasize frequently, rapidly, and most often within the peritoneum or to local lymph nodes, but also to distant sites (lung, liver, spleen) [8,11,13,15]. Subtotal colectomy via laparotomy is the current standard treatment but mean overall survival time of cat patients is generally low (68 to 274 days), as recurrence and metastases are common [11,15,16].

The aim of the present study was to characterize sporadic primary feline intestinal carcinomas histologically and molecularly in order to further compare them to human CRC. WHO subtype, tumor grading and tumor budding status are valuable prognostic tools in human CRC [17,18] and feline tumors were characterized according to the current human WHO classification [19].

Mutations, which stabilize the *CTNNB1* gene encoding for β-catenin and thus activate the canonical WNT/β-catenin signaling pathway, play a pivotal role in the pathogenesis of human [20,21] and canine [22] intestinal cancer. Based on previous immunohistochemical findings of other authors, showing that dysregulated and nuclear translocated β-catenin was present in spontaneous feline intestinal carcinomas [12], we shed light on the mutational status of feline *CTNNB1* by performing Sanger sequencing of feline *CTNNB1* exon 2 for the first time.

## 2. Materials and Methods

### 2.1. Feline Study Cohort

Thirty-three cases of spontaneous feline colorectal and sixteen cases of feline small intestinal carcinoma were collected from the tissue archive of LABOKLIN GmbH & Co. KG (Bad Kissingen, Germany) between the years 2013 and 2020. All samples (7 full-thickness biopsies and 42 surgical specimens) were obtained during laparotomy, submitted for pathological routine diagnostics, and reviewed retrospectively. In 20/49 cases, additional lymph node samples were available.

Intestinal tumor samples originated from cats of different breeds including 31 European Shorthair (ESH) cats (63.3%), 4 British Shorthair cats (8.2%), 3 Persian cats (6.1%), 2 Chartreux cats (4.1%), 2 Maine Coon cats (4.1%), 1 Exotic Shorthair cat (2%), 1 Norwegian Forest cat (2%), 1 Oriental Shorthair cat (2%), 1 Siamese cat (2%) and 3 mixed cat breeds (6.2%). The study set included 26 males (53.1%; 20/26 castrated) and 23 females (46.9%; 15/23 spayed), and the age at the time of diagnosis ranged from 4 to 17 years. Mean age (± SD) was 11.51 years (± 3.31 years). Detailed information on the feline cohort is provided in Appendix A.

#### 2.1.1. Tissue Processing

Tissue samples were fixed in 10% neutral-buffered formalin and routinely processed for histology. Hematoxylin and eosin (H&E) staining was carried out following standard protocols. A Periodic Acid Schiff (PAS) reaction was performed on six selected cases in order to visualize intracytoplasmic mucin for validating the diagnosis of signet-ring cell carcinoma (SRCC). All slides were scanned in 40× magnification using a high-throughput slide scanner (Aperio AT2, Leica Biosystems, Wetzlar, Germany). The histological classification of tumors was based on H&E and PAS-stained slides and carried out by a trainee veterinary pathologist (T.G.) under the supervision of experienced board-certified human (M.J., C.M.) and veterinary pathologists (D.D., K.S.). Immunohistochemistry (IHC) for β-catenin, Ki-67, Pan-cytokeratin and CD31 was performed. Detailed information on the IHC protocols and primary antibodies is provided in Appendix A. Representative images were taken using Aperio ImageScope ×64 (v.12.4.0.7018, Leica Biosystems, Wetzlar, Germany).

#### 2.1.2. Histomorphological Characterization

For comparative reasons, the histological subtype classification of feline tumors was performed based on the current human WHO classification guidelines [19]. The following human WHO subtypes were identifiable amongst the feline intestinal carcinomas: adenocarcinoma not otherwise specified (ANOS), serrated adenocarcinoma (SAC), mucinous adenocarcinoma (MAC), micropapillary adenocarcinoma (MPC), and signet-ring cell carcinoma (SRCC). ANOS is considered a malignant epithelial neoplasia displaying glandular differentiation. SAC is characterized by glandular serration and consists of tumor cells with a low nucleus-to-cytoplasm ratio, which may be mixed with mucinous areas. MAC is defined by significant pools of extracellular mucin that contain tumor cells and form >50% of the tumor. MPC consists of ≥5% of small tumor cell clusters surrounded by stromal spaces, morphologically mimicking lymphatic or vascular channels, and therefore displays typical retraction artefacts [23]. SRCC consists of signet-ring cells forming >50% of the tumor, and containing prominent intracytoplasmic mucin, characteristically impressing and partially displacing the nucleus to the periphery. Carcinomas of all types with <50% areas containing mucin are designated as having a mucinous component [19].

The International Tumor Budding Consensus Conference (ITBCC) of 2016 achieved standardized classification for tumor budding in human colon cancer, establishing a clearly delineated tumor budding scoring scheme. Tumor buds were defined as individual or clusters of up to four cancer cells detached from the main tumor mass and counted in one hotspot area (0.785 mm^2^) at the invasive front on an H&E slide (20×). Clinically relevant cut-off values were defined in a 3-tier system as low (0–4 buds), intermediate (5–9 buds) and high (≥10 buds), and termed Bd1, Bd2, and Bd3, respectively [18] (Appendix A). Tumor budding assessment was carried out for feline tumors in the same manner (Figure 1).

Adapted from the human WHO classification of 2019, a 2-tiered grading of feline intestinal tumor was based on the differentiation degree of cellular gland formations in the least differentiated tumor area. In consequence, the neoplasms were categorized as either “low-grade” (≥50% gland formation; well to moderately differentiated) or “high-grade” (<50% gland formation; poorly differentiated) [19].

All neoplasms were scored for 10 additional histological parameters including vascular and lymphatic invasion, perineural growth, invasion depth, lymph node metastasis, inflammatory cell infiltration, scirrhous reaction, presence of osseus metaplasia, a mucinous component and mucosal ulceration. Vascular (extra- and intra-mural) and lymphatic invasion as well as perineural growth were assessed as either absent (0) or present (1). Invasion depth was scored as infiltrating the lamina muscularis propria (1), the tunica muscularis (2), or the serosa or greater omentum (3). A cumulative score of invasiveness, including vascular (1), lymphatic (1), perineural (1), and serosal (1) infiltration was calculated (max. score of 4). Regional lymph nodes were available for histological evaluation of metastasis in 20/49 cases. Cellular immune response was measured semi-quantitatively by scoring the inflammatory infiltrate in the tumorous area as either absent (0), mild (1; mucosal), moderate (2; mucosal, submucosal and partly involving the tunica muscularis), or severe (3; involving all intestinal layers). A scirrhous reaction, mucinous component, osseous metaplasia, and mucosal ulceration were determined as either absent (0) or present (1).

Mitotic count (MC) was assessed digitally in an area equaling 10 high-power fields (hpf; 40×) on a standard monitor using Aperio ImageScope X64 (Leica Biosystems). Assuming 2.37 mm^2^ was agreed to be the standard field area of 10 hpf and 0.0954 mm^2^ was the area of one hpf in the aforementioned setting, 25 40× image fields were counted to equal the standard hpf area. The MC was performed randomly within the most densely cellular areas of the neoplasm and cell poor areas were excluded [24]. The total number of mitoses/10 hpf (2.37 mm^2^) were scored as follows: 0–9 mitoses (1); 10–19 mitoses (2); and ≥20 mitoses (3).

#### 2.1.3. Semiquantitative Evaluation and Computer-Assisted Image Analysis

Beta-catenin immunoreactivity, in terms of a nuclear translocation of β-catenin, was scored semiquantitatively: 0 (negative; <5% positive cells); 1 (5–25% positive cells); 2 (26–50% positive cells); and 3 (>50% positive cells) (modified score from Uneyama et al.) [12] (Figure 2).

Regarding proliferative activity, the Ki-67 index was assessed by a computer assisted algorithm. Selected regions of interest (ROI), i.e., tumor areas, were manually annotated by a trainee pathologist (T.G.). The ROI were exported as xml files and transferred into an open-source image analysis software (‘QuPath version 0.2.3, https://qupath.github.io, University of Edinburgh, Scotland) for quantification. The default set of parameters of the algorithm was modified according to the stain contrast and intensity of the scanned images. Cell segmentation was performed using the following settings: detection image, optical density sum; requested pixel size 0.5 µm; background radius 8 µm; median filter radius 1 µm; sigma 1.5 µm; minimum cell area 10 µm^2^; maximum cell area 400 µm^2^, threshold 0.1; and maximum background intensity 2. Cell classification (tumor cells, immune or stromal cells) was completed after training an object classifier using ‘Random trees’ as a machine learning method [25]. ‘Smoothed object features’ at a 25 µm radius were added. This was to help with segmenting an image homogeneously so that the classifier performed an accurate classification. As a quality control step, the results of segmentation and correct cell classification were reviewed by a trainee pathologist (T.G.). Finally, the Ki-67 proliferation index was calculated exclusively within the class “tumor cells” as the percentage of cells with positive Ki-67 immunostaining and was scored as follows: <5% (0); 5–30% (1); 31–50% (2); 51–80% (3); and >80% (4).

#### 2.1.4. Sanger Sequencing of Feline *CTNNB1* Exon 2

In order to elucidate the relevance of feline *CTNNB1* exon 2 alterations in intestinal carcinogenesis we established a Sanger sequencing protocol for feline *CTNNB1* gene exon 2 encoding β-catenin. In a first step, we compared and aligned the human (ENST00000349496.11, NM_001904.4, hg19) and feline DNA-sequences of *CTNNB1* (ENSFCAT00000003470.6, Felis_catus_9.0) to identify the corresponding regions of interest in the feline sequence. Human *CTNNB1* exon 3 is known to include a hotspot region of frequent mutations in various cancer entities, e.g., liver, stomach, and colorectal cancer [26]. Therefore, by comparing the two sequences, we identified feline *CTNNB1* exon 2 as homologous to the human nucleotide sequence of exon 3. According to this, a specific primer pair, 5′-AGCTGATCTGATGGAACTGGAC-3′ (forward) and 5′-ACACCCTTACCAGCCACTTG-3′ (reverse), which amplifies a 237-bp product encompassing feline *CTNNB1* exon 2, was designed. This primer pair was previously established and validated in a pre-study of feline fibrosarcoma (*n* = 5) using tumor samples and matching normal tissue samples. In brief, the DNA was extracted from areas of interest on FFPE sections (tumor tissue and/or normal tissue) by means of a Maxwell^®^ RSC Blood DNA Kit (Promega, Madison, WI, USA) according to the manufacturer’s protocol. DNA concentration was afterwards fluorimetrically measured by using a Qubit 4.0 system and the Qubit DNA high sensitivity Assay (both: Thermo Fisher Scientific, Waltham, MA, USA). For amplification of the region of interest (feline *CTNNB1* gene, exon 2) 10–20 ng of DNA was used as input for the polymerase chain reaction (PCR). PCR was performed with AmpliTaq Gold polymerase (Thermo Fisher Scientific, Waltham, MA, USA) in an Eppendorf Mastercycler^®^ Gradient (Eppendorf AG, Hamburg, Germany) with an annealing temperature of 60 °C. Subsequently, the amplification of tumor samples and negative control (non-template control) was validated using agarose gel electrophoresis and visualization on an Amersham Imager 680 detection system (General Electric Company Healthcare Bio-Sciences AB, Uppsala, Sweden). For purification, the PCR products were digested using ExoSAP nuclease (New England Biolabs, Ipswich, MA, USA) for 15 min at 37 °C followed by inhibition of the enzyme at 80 °C for 10 min in an Eppendorf Mastercycler^®^. Sanger sequencing was performed using the BigDye v1.1 Terminator Mix (Thermo Fisher Scientific, Waltham, MA, USA) according to the manufacturer’s protocol in an Eppendorf Mastercycler^®^. For capillary electrophoresis the sequencing product was purified using the ZR DNA Sequencing Clean-up Kit^TM^ (Zymo Research Europe GmbH, Freiburg, Germany) and loaded on an ABS/Hitachi 3130 genetic analyzer (Thermo Fisher Scientific, Waltham, MA, USA). After sequencing, electropherograms of each tumor sample and corresponding normal tissue were visually analyzed for the occurrence of mutations.

In this study, eleven cases positive for nuclear β-catenin (score 1–3) and containing a sufficient quantity of tumor cells (>30% tumor cell content) were selected for molecular analysis of the feline β-catenin gene exon 2 according to the method described above. Healthy intestinal mucosa from the same animals (*n* = 7) and from other animals (*n* = 2) was used as a negative control in order to confirm the native sequence of feline *CTNNB1*.

### 2.2. Human Specimen

For comparative purposes, human CRC specimens from the diagnostic archive of the Institute of Pathology of the Technical University of Munich were evaluated. The use of human tissue was approved by the local ethics committee of the Technical University of Munich/Klinikum Rechts der Isar (reference number: 506/17 s).

### 2.3. Statistical Analysis

Statistical analyses were performed using SPSS software version 27.0.1.0 (SPSS Institute, Chicago, IL, USA). Associations between more than two samples (i.e., WHO subtypes) and the assessed histological features (grade; cumulative score of invasiveness; vascular, lymphatic, and perineural invasion; lymph node metastasis; tumor budding; inflammation; scirrhous reaction; mucinous component; osseous metaplasia; mucosal ulceration; proliferation (MC, Ki-67); and β-catenin translocation) were examined via a Kruskal–Wallis test for nonnormally distributed parameters and a Bonferroni-adjusted post-hoc analysis. Trends between two samples (i.e., grades, tumor localization) were tested via a Mann–Whitney U test. A *p* value < 0.05 was considered statistically significant for all data sets.

## 3. Results

### 3.1. Tumor Site and Frequency

Histomorphological evaluation showed that the majority of examined feline intestinal carcinoma (33/49 cases, 67.3%) was located in the large intestine, whereas 16/49 cases (32.7%) appeared in the small intestine. Due to the striking morphologic similarity with human intestinal neoplasias, we decided to also include small intestinal tumors in the feline study set. In humans, small intestinal neoplasms are rare compared to colonic adenocarcinoma but the subtypes resemble the colonic classification [27].

### 3.2. Distribution of Histopathological Subtypes of the Feline Intestinal Tumor Cohort

Feline intestinal carcinomas closely resembled human WHO subtypes on a histomorphological level (Figure 3). Of the 16 small intestinal tumors, 12 were classified as ANOS (75%), two as MAC (12.5%), one as SAC (6.3%) and one as MPC (6.3%). Of the 33 tumors of the colon and rectum, 17 were classified as ANOS (51.5%), 12 as SAC (36.4%), 3 as MAC (9.1%), and 1 as SRCC (3%). Overall, ANOS was the most common histological subtype in cats (59.2%) followed by SAC (26.5%). MAC comprised 10.2%, and MPC and SRCC 2% of cases, respectively (Table 1). Other human WHO subtypes, including adenoma-like adenocarcinoma, medullary adenocarcinoma, adenosquamous carcinoma, undifferentiated carcinoma, and carcinoma with sarcomatoid components, were not identified in the investigated feline tumor set.

### 3.3. Histopathological Features of Feline Intestinal Tumor Subtypes

The majority of small intestinal carcinomas were of high grade (14/16; 87.5%). A mucinous component was found in 10/16 cases (62.5%). Vascular invasion was present in 5/16 cases (31.3%), perineural invasion in 3/16 cases (18.8%) and lymphatic invasion in 9/16 cases (56.3%). Mesenterial lymph nodes were available in 5/16 cases and metastasis was present in 2 of those 5 cases. Serosal infiltration was present in 8/16 cases (50%). Invasiveness was high (score 3) in 5/16 cases (31.3%) and low (score 0) in 3/16 cases (18.8%). Inflammation was mild in 12.5%, moderate in 50% and severe in 37.5% of cases. Mucosal ulceration was present in 13/16 cases (81.3%). A scirrhous reaction was found in 11/16 cases (68.8%). No osseous metaplasia was present. The number of mitotic figures ranged from 1 to 17, mean (± SD) number of mitoses was 6.88 (± 4.573) and the median was 5.5. All small intestinal carcinomas were of the highest budding grade (Bd3).

Regarding colonic carcinomas, 54.5% were of low grade and 45.5% of high grade. A mucinous component was identified in 22/33 cases (66.7%). Vascular invasion was present in 13/33 cases (39.4%), perineural invasion in 13/33 cases (39.4%) and lymphatic invasion in 11/33 cases (33.3%). Lymph node metastasis was present in 11 of 15 (73.3%) submitted lymph node samples. Inflammation was mild in 60.6% (20/33), moderate in 15.2% (5/33) and severe in 24.2% (8/33) of the colonic carcinomas. The majority of colonic carcinomas had a scirrhous component (28/33; 84.8%) and osseous metaplasia was present in 7/33 cases (21.2%). The number of mitoses per 10 hpf ranged from 0 to 37, the mean number was 7.42 (± 7.87), and the median was 6. The tumor budding status of colonic carcinomas was generally high (84.3% Bd3).

Overall, 20 intestinal tumors were classified as low grade (40.2%) and 29 as high grade (59.8%). Low grade tumors grew significantly less invasive than high grade tumors (*p* = 0.025). A mucinous component was present in 32/49 cases (65.3%) of the investigated neoplasms. Vascular invasion was present in 36.7% *(p* = 0.137), perineural invasion in 32.7% (*p* = 0.527) and lymphatic invasion in 40.8% (*p* = 0.141) of all intestinal carcinomas. Metastases were present in 13 of 20 cases with available lymph nodes. Intestinal tumors penetrated the serosa in 27/49 cases. Mucosal ulceration was present in 37/49 cases (75.5%) (Figure 4). In 44.9% (22/49) of cases inflammation of the tumor area was mild and composed of a mixed inflammatory cell infiltrate. A scirrhous reaction was present in 39/49 (79.6%) and osseous metaplasia in 7/49 (14.3%) cases. In general, the feline tumor budding status was remarkably high (44/49 Bd3, 89.8%). Bd3 tumors had a significantly higher cumulative score of invasiveness (*p* = 0.006) and invaded blood (*p* = 0.205) and lymphatic vessels (*p* = 0.152) more frequently. For the overall cohort, statistical analysis revealed a significant difference between the WHO subtypes regarding the feature invasiveness, represented by a cumulative score of invasiveness (*p* = 0.021). Feline serrated adenocarcinomas grew significantly less invasive than ANOS (*p* = 0.014) (Figure 5). Concordantly, ANOS infiltrated upon the serosa more frequently than SAC (*p* = 0.028). Strikingly, feline intestinal carcinomas generally exhibited an extremely dissociative and aggressively infiltrative growth pattern. For all the other aforementioned criteria no significant trends with regard to the specific histological subtypes could be determined. Detailed information on the relation between histological subtypes and the assessed histological and molecular features of the overall cohort is provided in Appendix A.

### 3.4. Immunohistochemical Features of the Feline Intestinal Tumor Cohort

Of the 16 small intestinal tumors, 93.8% showed no or scattered nuclear translocation of β-catenin (score 0). Nuclear translocation in 5–25% of tumor cells (score 1) was present in one small intestinal ANOS (6.3%). The 33 colonic carcinomas were positive for nuclear β-catenin to various degrees (69.7% score 0; 24.4% score 1; 3% score 3 or 4, respectively). Out of eight colonic tumors with a score of one, five were of the subtype ANOS, two were SAC and one was MAC. One colonic SAC was a score 2 and one colonic ANOS was a score 3. Taken together, the majority of the examined feline tumors displayed no or scattered nuclear translocation of β-catenin (score 0; 38/49 cases; 77.6%). Nine tumors were a score 1 (18.4%), one colonic tumor a score 2 (2%) and one a score of 3, respectively.

Tumor proliferation (Ki-67) did not differ statistically significant between the histological subtypes (*p* = 0.359) or grades (*p* = 0.26).

### 3.5. β-Catenin Gene Mutations in Exon 2 of Feline CTNNB1

For 11 tumor samples, which immunohistochemically displayed nuclear translocation of β-catenin (score 1, 2 or 3), Sanger sequencing identified somatic missense mutations in exon 2 of the feline *CTNNB1* gene in 4 colonic tumors (case 14: colonic ANOS score 1; case 20: colonic ANOS score 3; case 22: colonic ANOS score 1; case 31 colonic SAC score 1; 36.4% of all samples). Three of these four mutations were of somatic origin, and in none of the samples germline mutations were detected in exon 2 of the *CTNNB1* gene. In case 20, no matching physiological tissue of the same animal was available, thus, it could not be determined if the mutation was of somatic or of germline origin in this specific case. All identified mutations were exclusively heterozygous, and the mutational spectrum comprised a p.S37F (c.110C>T) (case 14), a p.D32G (c.95A>G) (case 20), a p.D32N (c.94G>A) (case 22), and a p.G34R (c.100G>A) (case 31) mutation (Figure 6). For all of these, an orthologous mutation is known in various human cancers, e.g., CRC, liver and stomach cancer [26]. In the remaining seven samples, a wildtype sequence of *CTNNB1* was identified either exclusively in the tumor tissue (*n* = 3) or in the tumor tissue and normal tissue (*n* = 4). Amplification status of *CTNNB1* could not be determined by Sanger sequencing.

### 3.6. Comparison of Feline Small Intestinal and Colonic Neoplasias

Small intestinal carcinomas were more often of high grade than colonic carcinomas (*p* = 0.005). Osseous metaplasia was present in the colonic, but not in the small intestinal carcinomas (*p* = 0.049). Nuclear translocation of the β-catenin was proved in one small intestinal ANOS (score 1), but visible to various degrees in 10 colonic tumors (*p* = 0.058). Somatic mutations of *CTNNB1* were exclusively detected in colonic carcinomas (*p* = 0.15). Inflammatory cell infiltration of the neoplastic area appeared to be mild in the majority of colonic cases and moderate to severe in the majority of small intestinal cases (*p* = 0.12). For all other assessed criteria, no statistically significant differences between the small intestinal and colonic carcinomas could be determined.

## 4. Discussion

The most striking morphological feature of the investigated and described feline cohort was an extremely high tumor budding activity related to a markedly dissociative tumor growth. Since the WHO criteria for human CRC was reclassified in 2019 and tumor budding was added, it is now recognized as a major grading criterion and a highly relevant and independent prognostic factor [17,19] that is generally considered to be a stage-independent predictor of lymph node metastasis in pT1 CRC and of survival in stage II CRC [18]. Tumor budding strongly impacts on all survival parameters and regarding its prognostic significance, it even outperforms WHO grade [17]. This study demonstrates a high budding status in feline intestinal carcinomas (89.8% Bd3) compared to human CRC (20% Bd3) [17]. To the authors’ knowledge, no other veterinary studies evaluate feline tumor budding, and further research to establish prognostic data is imperative.

Although many GEMMs are available, researchers frequently face the problem that induced intestinal neoplasms of mice lack the invasive features characteristic for late-stage CRC, e.g., tumor budding. There are orthotopic mouse xenograft models which show tumor budding that are morphologically and immunohistochemically close to what is seen in human CRC; however, these models often have an immunocompromised background and thus are limited in their relevance to the human situation [28,29]. Moreover, the very limited availability of in vivo budding models to date is another drawback [29]. Our study shows the capability of spontaneous feline intestinal carcinomas to serve as an immunocompetent model for elucidating the intestinal tumor budding mechanisms of CRC.

Due to these similarities and the potential use of spontaneously arising feline intestinal tumors for comparative research trials of human CRC, we decided on categorizing feline intestinal tumors according to the human WHO classification of 2019 [19]. Human WHO subtypes were proven to be clinically relevant with a strong impact on overall survival (OS), disease-free survival (DFS), and disease-specific survival (DSS) (*p* < 0.001) and a clear association with WHO grade and budding status. For example, MPC and SRCC are very aggressive subtypes connected to a poor survival prognosis, whereas SAC mostly does not invade perineural or venous and is connected to a better prognosis considering the CRC subtypes [17].

The feline cohort was compared to a large-scale cohort of human colorectal carcinomas recently characterized and published by Jesinghaus et al. in 2021. Most human CRCs are ANOS, defined by an invasive growth pattern breaking the line of the lamina muscularis mucosae and invading the submucosa; a feature which was present in all included feline intestinal tumors. Overall, ANOS was the most common histological subtype (59.2%) in cats, similar to the human cohort (62.7%) [17]. In a recent study of feline intestinal carcinomas, tubular adenocarcinoma was determined to be the most common tumor type (33/50 cases; 66%), morphologically comparable to ANOS [12]. The second most common colonic subtype in the feline cohort was SAC, which was overrepresented compared to human CRCs (8.7%) [17]. Consistent with this, Uneyama et al. found that feline colorectal carcinomas frequently showed glandular serration, and they detected three *KRAS* mutations in seven cases of feline colorectal epithelial tumors [12]. Presuming that *KRAS* gene mutations are frequently involved in human CRC development and particularly in the ‘serrated pathway’, this pathway may play an important role in feline intestinal carcinogenesis, as it does for human serrated adenocarcinomas [20,30,31,32]. Although we unfortunately lack survival data for our described feline study set, we were able to show that feline SAC displayed less invasive growth compared to other subtypes, compatible with the rather favorable prognosis of human SAC [17].

A lack of species-specific investigation tools, especially of molecular pathological markers, constitutes a significant challenge in the use of companion animal cancers as human tumor models (e.g., DNA primers). Commercially available and formalin-approved antibodies for cats or dogs are not as readily available as for human and rodent tissue [7] and molecular methods aiming at the detection of specific somatic or germline mutations are infrequently used in companion animal studies. We successfully designed a DNA primer pair appropriate for amplification of the feline *CTNNB1* gene exon 2, which is homologous to the nucleotide sequence of human *CTNNB1* exon 3 and contains a hotspot region of frequent mutations for various human cancers including CRC [26]. Because of this homology, *CTNNB1* mutations located in this DNA-region are most likely to cause similar effects in both species.

Mutations activating the canonical WNT/β-catenin signaling pathway are very frequently involved in the ‘classical pathway’ of human colorectal carcinogenesis [20]. β-catenin is a highly conserved protein, part of the WNT signaling pathway and plays an important role in cell-adhesion [33]. Alterations of *CTNNB1* that result in disturbed degradation of β-catenin lead to its cytoplasmic accumulation and subsequent translocation into the nucleus, where it acts as an oncogenic player enhancing the expression of several downstream target genes, e.g., *CCND1* (CYCLIN D1) [34,35], *MYC* [20,35,36], and *AXIN2* [35]. On the one hand, stabilizing homozygous *CTNNB1* mutations play a crucial role, especially in human CRC associated with Lynch syndrome [21]. On the other hand, *CTNNB1* mutations are less often (5% non-hypermutated (nHM) CRC; 7% hypermutated (HM) CRC) involved in sporadic human colorectal carcinogenesis than *APC* (Adenomatous polyposis coli) mutations (81% nHM; 53% HM CRC) [20].

From a comparative point of view, in intestinal neoplasms in dogs, *CTNNB1* mutations were proven to be more often causative than *APC* mutations, with *CTNNB1* being mutated in >60% of canine colorectal tumors [22]. Several studies also provide evidence of nuclear β-catenin translocation and accumulation in canine intestinal adenomas and carcinomas [37,38,39].

Currently, very little is known about the genomic landscape of companion animal cancers [40], and particularly of feline cancers. Because β-catenin score did not correlate with malignancy, Uneyama et al. concluded from their IHC results that dysregulated β-catenin is likely not an important player in feline intestinal tumorigenesis; however, accumulation of β-catenin was evident in 60% of their cases [12]. Our immunohistochemical examination of feline intestinal carcinomas (22.5% of cases positive for nuclear β-catenin) as well as the high expression of active β-catenin in feline mammary tumors compared to healthy tissue [41], prompted us to further investigate the role of this key protein in the entity of feline intestinal cancer.

For the first time, Sanger sequencing of feline *CTNNB1* exon 2 was performed and revealed four somatic missense mutations identical with pathogenic mutations in humans [26]. Human codons most frequently displaying mutations related to CRC are, namely, codons 32, 33, 34, 37, 41, and 45 [26]. Canine mutations were found in codons 32, 34, and 45 [22]. In our study, feline mutations were located at codons 32, 34 and 37, consistent with mutations of human and canine intestinal tumorigenesis. As a result, we strongly challenge the finding that dysregulated WNT/β-catenin signaling is not involved in feline intestinal carcinogenesis.

*APC* loss-of-function mutations leading to an impacted degradation of β-catenin can also lead to an increase of cytoplasmic and nuclear β-catenin [20]. This could be a possible explanation for the seven cases, which displayed nuclear β-catenin positivity but did not show sequential alterations of *CTNNB1*. Future investigations conducting feline *APC* sequencing (i.e., panel or whole exome sequencing) and including a larger cohort size are required to finally clarify if *APC* mutations also play a role in feline intestinal tumorigenesis. Nevertheless, genomic amplification of the *CTNNB1* gene, which might also be a mechanism for overexpression of the protein in cancer in humans, cannot completely be excluded [42].

The majority (93.8%) of small intestinal tumors displayed no relevant nuclear β-catenin translocation and the only tumor showing a nuclear IHC-signal did not harbor a *CTNNB1* mutation. In contrast to that, colonic tumors displayed various degrees of positivity for nuclear β-catenin (*n* = 10) and mutations of *CTNNB1* (*n* = 4). Although the feline small and large intestinal tumors appear to have histomorphological similarities, small intestinal tumors were almost exclusively of high grade and high tumor budding status. Future investigations are needed to further elucidate the mutational spectrum of feline small intestinal tumors.

A first step has been taken, but much work remains to be done. In order to beneficially integrate the feline model into human CRC research, further investigation of feline cancers’ genetics, genome-wide studies as well as genome annotation are imperative. The final ideal of comparative oncology is to include companion animals (pets) with comparative cancer diseases in clinical trials. Educating and informing pet owners and considering ethical standards is a major point here [40].

Our data provides an accurate histological classification system for feline intestinal tumors and a basis for comparative oncology [40] studies by harmonizing histological classification and conducting molecular examination on spontaneously arising intestinal tumors in pet cats. The results indicate that feline intestinal carcinomas constitute a valuable and promising in vivo model for human CRC, worthy of further characterization.

## 5. Conclusions

In conclusion, the present study evaluates histopathological features and patterns of feline intestinal tumors. It demonstrates two main reasons for the suitability of feline intestinal tumors as a valid spontaneous in vivo model for late-stage human CRC: (1) Feline intestinal carcinomas resemble human subtypes and present with an invasive growth and high tumor budding activity, (2) *CTNNB1* mutations are present in feline intestinal carcinomas, as has been reported in human and canine intestinal tumorigenesis. Sharing two important molecular alterations, namely *KRAS* [12] and *CTNNB1* mutations involved in intestinal carcinogenesis, cats are a valuable model for late-stage sporadic human CRC.

This study provides a solid foundation for the comparison of feline and human CRC, indicates the need to review the available classification schemes for feline intestinal cancers and paves the way for future comparative oncology studies.

## Figures and Tables

**Figure 1 cancers-13-05941-f001:**
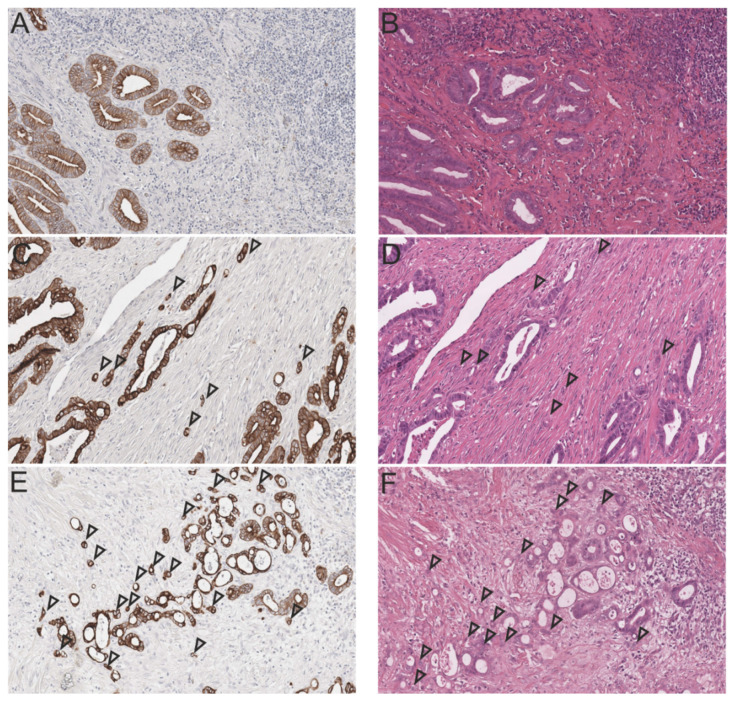
Tumor budding (TB) in the invasive front of feline intestinal carcinomas according to the 3-tier-system for budding assessment of human CRC (left: Pan-cytokeratin, right: H&E, consecutive sections, 20×). Tumor buds are indicated by arrow heads; (**A**) Low TB activity (Bd1, Pan-cytokeratin); (**B**) Low TB activity (Bd1, H&E); (**C**) Moderate TB activity (Bd2, Pan-cytokeratin); (**D**) Moderate TB activity (Bd2, H&E); (**E**) High TB activity (Bd3, Pan-cytokeratin); and (**F**) High TB activity (Bd3, H&E). For corresponding human H&E sections see Appendix A.

**Figure 2 cancers-13-05941-f002:**
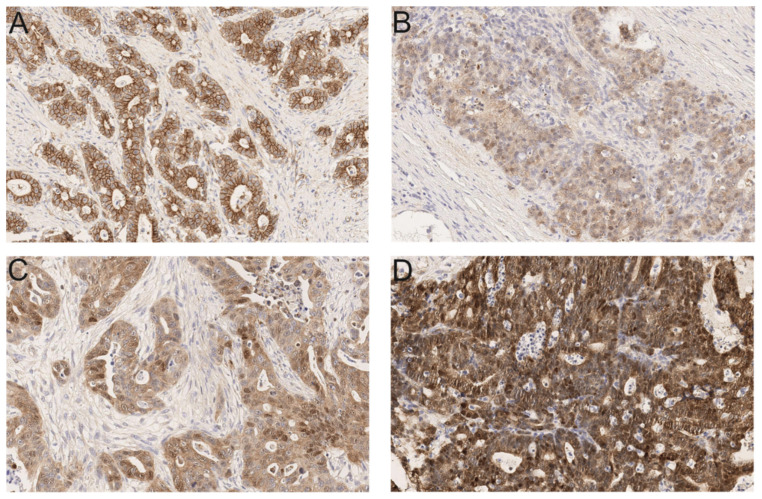
Scoring for nuclear translocated β-catenin in feline intestinal carcinomas (Anti-β-catenin; 20×). (**A**) Tumor with no or scattered (<5%) nuclear β-catenin staining (score 0); (**B**) 5–25% of tumor cells display nuclear positivity for β-catenin (score 1); (**C**) 26–50% of tumor cells display nuclear positivity (score 2); and (**D**) >50% of tumor cells are positive for nuclear β-catenin (score 3). For corresponding human β-catenin stainings see Appendix A.

**Figure 3 cancers-13-05941-f003:**
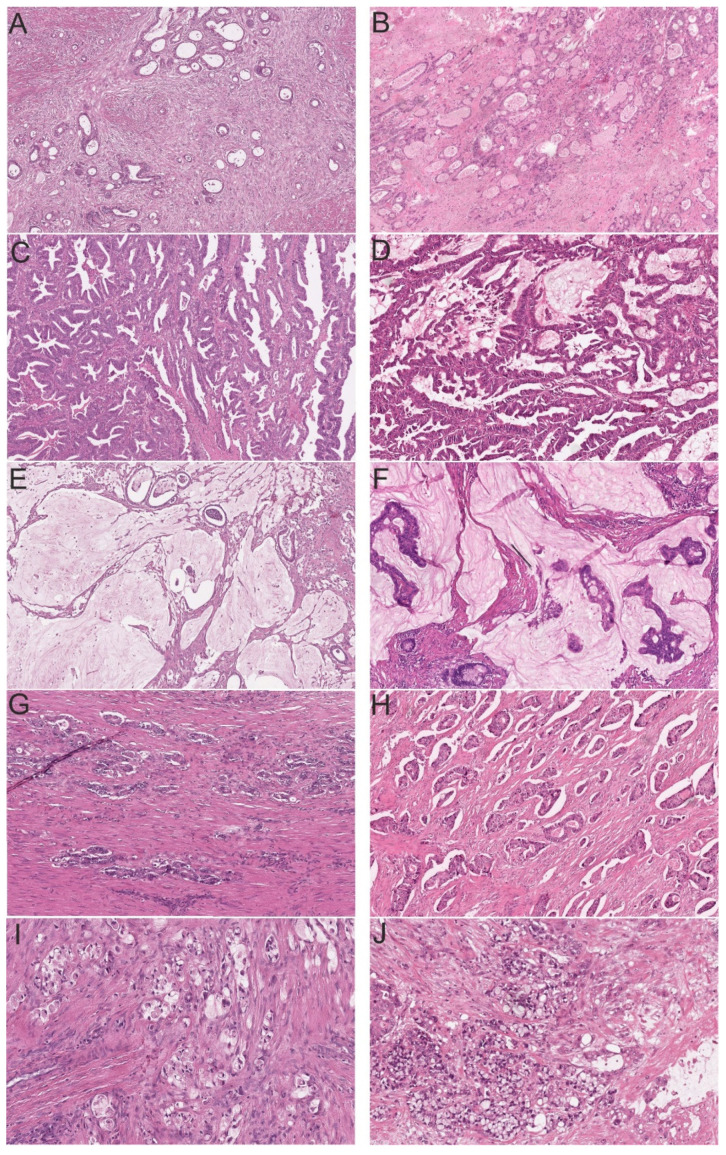
Feline intestinal carcinomas (**left**) closely resemble human WHO subtypes (**right**) (H&E). (**A**) Feline colonic adenocarcinoma not otherwise specified (ANOS, 8×); (**B**) Human colonic ANOS (8×); (**C**) Feline colonic serrated adenocarcinoma (SAC, 8×); (**D**) Human colonic SAC (8×); (**E**) Feline small intestinal mucinous adenocarcinoma (MAC, 8×); (**F**) Human colonic MAC (8×); (**G**) Feline small intestinal micropapillary carcinoma (MPC, 8×); (**H**) Human colonic MPC (8×); (**I**) Feline colonic signet-ring cell carcinoma (SRCC, 20×); and (**J**) Human colonic SRCC (20×).

**Figure 4 cancers-13-05941-f004:**
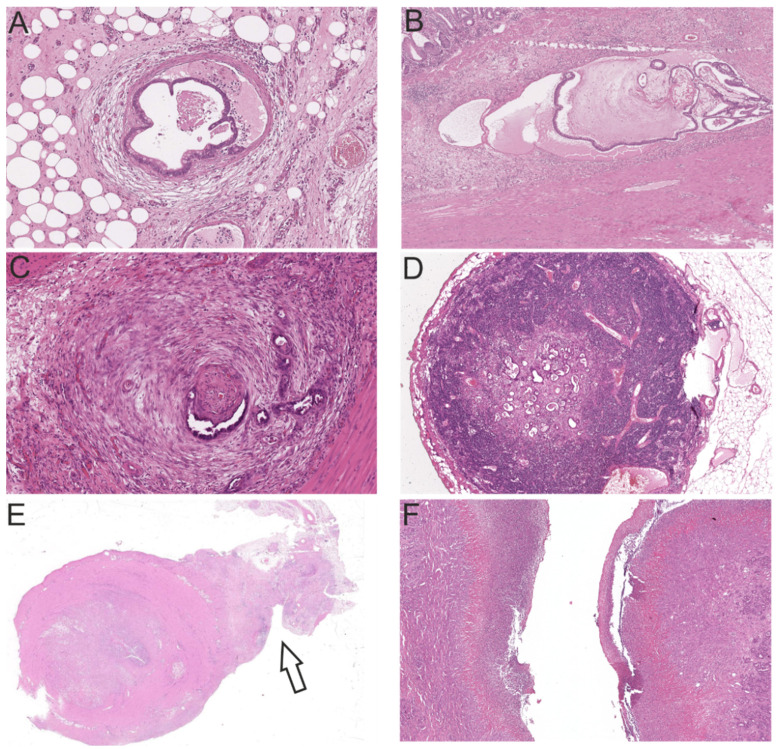
Characteristic malignancy features of feline intestinal tumors (H&E). (**A**) Vascular invasion (10×); (**B**) Lymphangiosis carcinomatosa (4×); (**C**) Perineural invasion (10×); (**D**) Lymph node metastasis (4×); (**E**) Serosal invasion, arrow (1×); and (**F**) Mucosal ulceration (4×).

**Figure 5 cancers-13-05941-f005:**
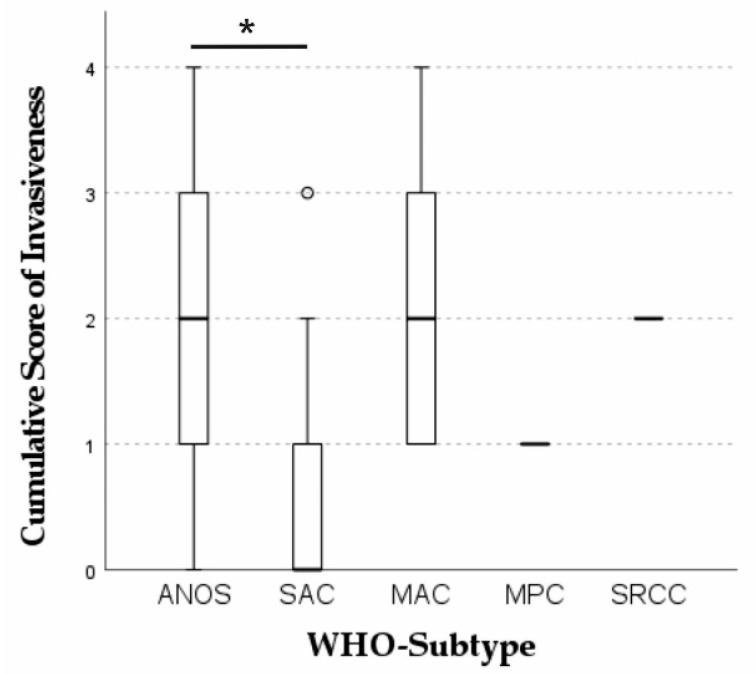
Kruskal–Wallis test of WHO subtypes regarding invasiveness. For the overall cohort, statistical analysis revealed a significant difference between the WHO subtypes regarding the feature invasiveness, represented by a cumulative score of invasiveness (vascular (1), perineural (1), lymphatic (1) and serosal invasion (1), max. score of 4). (*p* = 0.021). The score of invasiveness was significantly higher for ANOS than for SAC (* *p* = 0.014).

**Figure 6 cancers-13-05941-f006:**
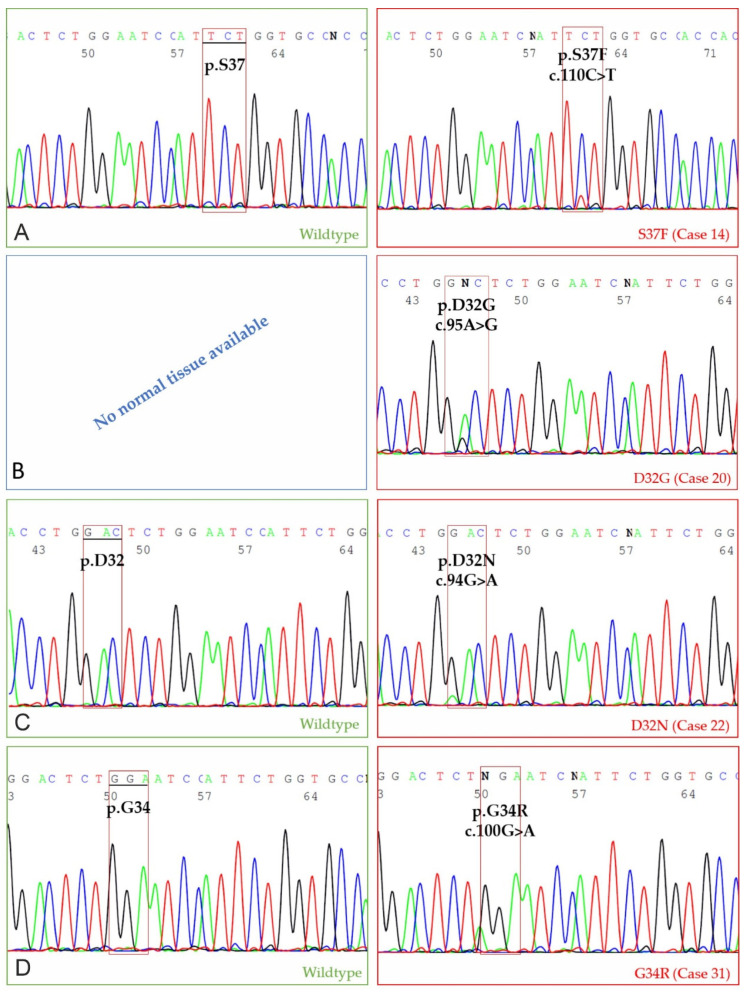
Four distinct somatic missense mutations were detected, each of them in a different case. DNA forward sequences; green: wildtype sequence of normal intestinal tissue (*n* = 3); red: tumor Appendix A. (**A**) Cytosine (C) is substituted by Thymine (T), resulting in a missense mutation leading to a replacement of Serine (S) by Phenylalanine (F) on the protein level (colonic ANOS; β-catenin score 1); (**B**) Adenosine (A) is substituted by Guanin (G) resulting in a missense mutation leading to a replacement of Aspartic acid (D) by Glycine (G) (colonic ANOS; score 3). For case 20, no normal tissue was available; (**C**) Guanin (G) is substituted by Adenosine (A) resulting in a replacement of Aspartic acid (D) by Asparagine (N) (colonic ANOS, score 1); (**D**) Guanin (G) is substituted by Adenosine (A) resulting in a replacement of Glycine (G) by Arginine (R) (colonic SAC, score 1).

**Table 1 cancers-13-05941-t001:** Distribution of histological WHO subtypes of feline intestinal carcinomas.

Cohort	Subtype	*n*	% of Total
Histological Subtypes (Overall Cohort, *n* = 49)	ANOS	29	59.18
	SAC	13	26.53
	MAC	5	10.20
	MPC	1	2.04
	SRCC	1	2.04
Histological Subtype (Small Intestinal, *n* = 16)	ANOS	12	75.00
	SAC	1	6.25
	MAC	2	12.5
	MPC	1	6.25
Histological Subtype (Colonic, *n* = 33)	ANOS	17	51.52
	SAC	12	36.36
	MAC	3	9.09
	SRCC	1	3.03

ANOS = adenocarcinoma not otherwise specified; SAC = serrated adenocarcinoma; MAC = mucinous adenocarcinoma; MPC = micropapillary carcinoma; SRCC = signet-ring cell carcinoma.

## Data Availability

The raw data of the results presented in this study are available on request from the corresponding author.

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
