# Peer review of "Bridging the Species Gap: Morphological and Molecular Comparison of Feline and Human Intestinal Carcinomas"

_cancers, 2021, doi:10.3390/cancers13235941_

Round 1

Reviewer 1 Report

Groll and colleagues have outlined an interesting dataset establishing the feline model of human CRC. The work is very detailed, complete and convincingly showcases the feline CRC model system, and its similarities with the human disease, in terms of morphology and some clinical parameters. The enhanced tumour budding phenotype is a particularly intriguing aspect of the feline model and may provide an interesting avenue to pursue.

Some considerations for the authors are as follows:

It is important to detail the drawbacks of the feline model system in order to provide a more balanced view of where and how the model system could be best applied – what issues are apparent for hypothesis testing: the lack of genetics, different molecular models, annotated genome, husbandry and ethical considerations, etc…

For statements such as ‘… strikingly similar to human CRC…’, a side-by-side comparison with corresponding, stained human sections, would enhance the point; for instance, for the molecular data in Figures 1-3.

Could the authors comment on the D32 and G34 CTNNB1 mutations – I don’t think these are hotspot mutations in human CRCs. Maybe gastric cancers?

Why were APC mutations not investigated?

Was the disproportionately high number of small intestinal epithelial tumours (relative to human intestinal epithelial cancers) related to a sampling bias?

Author Response

Groll and colleagues have outlined an interesting dataset establishing the feline model of human CRC. The work is very detailed, complete and convincingly showcases the feline CRC model system, and its similarities with the human disease, in terms of morphology and some clinical parameters. The enhanced tumour budding phenotype is a particularly intriguing aspect of the feline model and may provide an interesting avenue to pursue.
Response: We would like to thank the respected reviewer for these kind and supporting words.

Some considerations for the authors are as follows:

  • It is important to detail the drawbacks of the feline model system in order to provide a more balanced view of where and how the model system could be best applied – what issues are apparent for hypothesis testing: the lack of genetics, different molecular models, annotated genome, husbandry and ethical considerations, etc…
    Response: Thanks for this suggestion, which is great. Accordingly, we have added “A first step has been taken, bus much work remains to be done. In order to beneficially integrate the feline model into human CRC research, further investigation of feline cancers’ genetics, genome-wide studies as well as genome annotation are imperative. The final idea of comparative oncology is to include companion animals (pets) with comparative cancer disease in clinical trials. Educating and informing pet owners and considering ethical standards is a major point here [40]” to the discussion part (line 528-533).

  • For statements such as ‘… strikingly similar to human CRC…’, a side-by-side comparison with corresponding, stained human sections, would enhance the point; for instance, for the molecular data in Figures 1-3.
    Response: Thanks for this valuable advice. In order to better outline and show the morphological similarities of human and feline phenotypes we have added corresponding human H&E/ β-Catenin sections to Fig. 1 (Supplementary Fig.1), Fig. 2 (Supplementary Fig.2), and to Fig. 3 (now on the right side of Fig. 3; images were previously Supplementary Fig. 1). Due to a lack of space and to provide the figures in an appropriate size and resolution for the reader, we decided to add the corresponding human sections of Fig. 1 and 2 to Supplementary material.
  • Could the authors comment on the D32 and G34 CTNNB1 mutations – I don’t think these are hotspot mutations in human CRCs. Maybe gastric cancers?
    Response: Thank you for this remark. We clarified that not all of these specific mutations are hotspot mutations for human CRC but have been described as hotspots in various human cancers (line 472), e.g., liver and stomach cancers. We clarified by making the following changes:
    “Human CTNNB1 exon 3 is known to include a hotspot region of frequent mutations in various cancer entities, g., liver, stomach, and colorectal cancer [26]” (line 227-228)
    “For all of these, an orthologous mutation is known in various human cancers, e.g., CRC, liver and stomach cancers [26] (line 389-390)

  • Why were APC mutations not investigated?
    Response: This is a very good point. Unfortunately, the size of human APC exon 16 is around 8.7 kb with a coding region of approximately 6.5 kb and the feline APC gene is of comparable size (6.9 kb/ coding region of approx. 6.7 kb). Because there are no hotspots of APC (tumor suppressor gene), it would be necessary to investigate at least the whole exon 16. Additionally, mutational alterations could also be located in other exons. Thus, proper investigation by using Sanger sequencing is not possible with respect to APC. Nevertheless, we are currently planning further studies with a feline cancer panel including APC. Accordingly, we have made the following change to the manuscript:
    “Future investigations conducting feline APC sequencing (e., panel or whole exome sequencing) and including a larger cohort size are required to finally clarify if APC mutations also play a role in feline intestinal tumorigenesis” (line 515).
  • Was the disproportionately high number of small intestinal epithelial tumours (relative to human intestinal epithelial cancers) related to a sampling bias?
    Response: Thanks for the question. To the best of our knowledge, there is no sampling bias or difference in clinical presentation or onset regarding large and small intestinal tumors in cats.
    There are contradictory studies (line 78) regarding the most common affected intestinal site in cats, and neoplasms of the small intestine seem to occur more frequently in domestic animals compared to human patients. Some studies and veterinary literature state that in cats, small intestinal neoplasms in general occur more frequently than colonic neoplasms (Klopfleisch, Robert. Veterinary Oncology, 2016. and Kessler, Martin, Kleintieronkologie: Diagnose und Therapie von Tumorerkrankungen bei Hund und Katze, 2012.). This is especially true for intestinal lymphomas, whereas carcinomas are more often located in the large intestine, as was shown in two recent studies with a cohort size of n = 293 and n = 1.129 (Schwittlick U, B.S., Aupperle-Lellbach H. Vorkommen und Lokalisation von gastrointestinalen Neoplasien bei 293 Katzen. Kleintiermed 2020, 6, 250-253 and Rissetto, K.; Villamil, J.A.; Selting, K.A.; Tyler, J.; Henry, C.J. Recent trends in feline intestinal neoplasia: an epidemiologic study of 1,129 cases in the veterinary medical database from 1964 to 2004. J Am Anim Hosp Assoc 2011, 47, 28-36, doi:10.5326/jaaha-ms-5554).

Reviewer 2 Report

The authors have assessed 49 cases of feline intestinal carcinoma and graded them according to human WHO classification. They have then compared them to a large cohort of human CRCs. In addition the authors have performed sequencing of the feline CTNNB1 gene to look for mutations. It is wonderful to see alignment of the two species in terms of histological phenotyping and preliminary molecular phenotyping, so as to allow for a better understanding of intestinal carcinoma in felines and for comparative oncology studies.

This is a very well carried out study, with well-presented findings. As such I only have a few issues (as well as some minor comments).

  • Regarding sequencing of exon 2 CTNNB1. 11 cases of tumor were sequenced, yet only 7 had healthy/normal tissue from the same animal (as detailed in lines 241-243). Thus it is only possible to be sure that these mutations are somatic (and not germline) for these 7 cases, not all 11 cases. This needs to be made clearer in the results section (Section 3.5).
  • Why was PAS reaction only performed on selected slides and not all slides? Perhaps indicate which cases had PAS performed on them?
  • Can the authors provide some explanation for the differences in their findings to that reported by Uneyama et al. in terms of the role of b-catenin in feline intestinal tumorigenesis?

Minor points

  • Line 25: I don’t feel it is correct to say that “Feline intestinal tumors perfectly model invasive late-stage human CRC”. They may represent a good or valid model, but not a “perfect” model.
  • Line 33: This sentence needs re-wording - “We investigated the role of the feline CTNNB1 gene by Sanger sequencing of exon 2”. Sequencing of an exon of a gene doesn’t constitute investigating the role of the gene. I would also move this sentence to later on in the abstract, so it appears just before the sentence starting “In four cases of feline colonic malignancies…..”
  • Line 44: change “Results” to “These results”.
  • Line 45: I don't think based on the analysis in this manuscript it is possible to conclude that these feline tumors constitute an “excellent” model. More analysis (particularly molecular pathology) is needed. Consider toning down the phrasing.
  • Line 65: Change “men” to “humans”.
  • Line 70: Remove “localizations,” and replace with “the”
  • Line 218: change “Brief” to “brief”
  • Line 275: can the panels in Figure 3 be arranged better?
  • Lines 395-397: the authors should make the point that whilst orthotopic mouse xenograft models exist, they are often on an immunocompromised background, and thus are limited in their relevance to the human situation.
  • Line 462: “For the first time, Sanger sequencing of feline CTNNB1 was performed”. Needs to be made clear that only exon 2 of the gene was sequenced.
  • Line 468: change “thesis,” to “finding”.
  • Line 484: the numbers used in this study are too small to be able to conclude that small and large intestinal tumors “seem to be different on a genomic level with respect to CTNNB1 mutations” (since only 1 small intestinal tumor was sequenced). This statement should be removed.
  • Line 497: “Dysregulated β-catenin due to CTNNB1 mutations seem to play a role in feline tumorigenesis, comparable with human and canine intestinal tumorigenesis”. This sentence needs to be reworded as “seem to play a role” is not scientific terminology. Perhaps something like “CTNNB1 mutations are found in feline intestinal carcinomas, as has been reported in human and canine intestinal tumorigenesis”.

Author Response

The authors have assessed 49 cases of feline intestinal carcinoma and graded them according to human WHO classification. They have then compared them to a large cohort of human CRCs. In addition the authors have performed sequencing of the feline CTNNB1 gene to look for mutations. It is wonderful to see alignment of the two species in terms of histological phenotyping and preliminary molecular phenotyping, so as to allow for a better understanding of intestinal carcinoma in felines and for comparative oncology studies.
This is a very well carried out study, with well-presented findings. As such I only have a few issues (as well as some minor comments).

Response: We would like to thank the respected reviewer for these kind and supporting words, the feedback on our study and the constructive comments.

  • Regarding sequencing of exon 2 CTNNB1. 11 cases of tumor were sequenced, yet only 7 had healthy/normal tissue from the same animal (as detailed in lines 241-243). Thus it is only possible to be sure that these mutations are somatic (and not germline) for these 7 cases, not all 11 cases. This needs to be made clearer in the results section (Section 3.5).
    Response: It is true, that this was not clearly stated. We clarified the results (section 3.5) accordingly:
    “In the remaining 7 samples, wild type sequence of CTNNB1 was identified either exclusively in tumor tissue (n = 3) or in tumor and normal tissue (n = 4).” (line 391-393)

  • Why was PAS reaction only performed on selected slides and not all slides? Perhaps indicate which cases had PAS performed on them?
    Response: Thank you for this comment. We have added this information accordingly:
    “Periodic Acid Schiff (PAS) reaction was performed on six selected cases in order to visualize intracytoplasmic mucin for validating the diagnosis of Signet-Ring Cell Carcinoma (SRCC)” to the paragraph 2.1.1. (Tissue Processing) (line 118-120).

  • Can the authors provide some explanation for the differences in their findings to that reported by Uneyama et al. in terms of the role of b-catenin in feline intestinal tumorigenesis?
    Response: The approach of Uneyama et al. was slightly different to ours, since they have investigated as well as compared feline adenomas vs. adenocarcinomas, and assessed features (e.g., nuclear β-catenin) were correlated to malignancy. Although they detected nuclear accumulation of β-catenin in 60% of their cases (5 adenomas, 25 adenocarcinomas), the authors did not pursue this avenue, because 48% of cases displayed just scattered nuclear positivity and their statistical results indicated, that there was no correlation of β-catenin score and malignancy. However, we thought, that this was an interesting avenue to pursue, and because 11 of 49 adenocarcinomas in our cohort displayed nuclear positivity and the study of Wang et al. 2018 intriguingly showed, that CTNNB1 was very frequently mutated in canine tumorigenesis, we aimed at having a look beyond IHC. We have added these facts starting at line 494:
    Because β-catenin score did not correlate with malignancy, Uneyama et al. concluded from their IHC results that dysregulated β-catenin is likely not an important player in feline intestinal tumorigenesis. However, accumulation of β-catenin was evident in 60% of their cases [12]. Our immunohistochemical examination of feline intestinal carcinomas (22.5% cases positive for nuclear β-catenin) as well as the high expression of active β-catenin in feline mammary tumors compared to healthy tissue [41] prompted us to further investigate the role of this key protein in the entity of feline intestinal cancer.”

  Minor points

  • Line 25: I don’t feel it is correct to say that “Feline intestinal tumors perfectly model invasive late-stage human CRC”. They may represent a good or valid model, but not a “perfect” model.
    Response: Thanks a lot for this reasonable remark. We have toned down the phrasing in line 26 (to “adequately”), line 50 (to “useful”) as well as in line 541 (to “valid”) and line 548 (to “valuable”).

  • Line 33: This sentence needs re-wording - “We investigated the role of the feline CTNNB1 gene by Sanger sequencing of exon 2”. Sequencing of an exon of a gene doesn’t constitute investigating the role of the gene. I would also move this sentence to later on in the abstract, so it appears just before the sentence starting “In four cases of feline colonic malignancies…..”
    Response: We thank the reviewer a lot for this important and mindful comment. The sentence has been moved to line 45 and changed to “We also investigated the relevance of feline CTNNB1 exon 2 alterations by Sanger sequencing.

  • Line 44: change “Results” to “These results”.
    Response: Thank you very much for this comment. We have changed this sentence according to the reviewer’s suggestion (Line 50).

  • Line 45: I don't think based on the analysis in this manuscript it is possible to conclude that these feline tumors constitute an “excellent” model. More analysis (particularly molecular pathology) is needed. Consider toning down the phrasing.
    Response: See point 1.

  • Line 65: Change “men” to “humans”.
    Response: Thanks for this comment. We have changed the wording according to the reviewer’s suggestion (line 71).

  • Line 70: Remove “localizations,” and replace with “the”
    Response: Thanks for the careful read. Correction made accordingly (line 76).

  • Line 218: change “Brief” to “brief”
    Response: Thank you for checking thoroughly. We have corrected the capital letter.

  • Line 275: Can the panels in Figure 3 be arranged better?
    Response: Thanks for this suggestion. In order to better showcase the similarities of human and feline phenotypes (suggestion by reviewer 1), we have changed Fig. 3 and added the corresponding human WHO subtypes (H&E) on the right side of the figure.

  • Lines 395-397: the authors should make the point that whilst orthotopic mouse xenograft models exist, they are often on an immunocompromised background, and thus are limited in their relevance to the human situation.
    Response: Thanks a lot for this great suggestion! We have added this crucial point to the second paragraph of the discussion section:
    “(…) models often have an immunocompromised background and thus are limited in their relevance to the human situation [28, 29]. Moreover, the very limited availability of in vivo budding models to date is another drawback [29].” (line 432-435) Additionally, we added that cats are an “immunocompetent model” (line 436).

  • Line 462: “For the first time, Sanger sequencing of feline CTNNB1 was performed”. Needs to be made clear that only exon 2 of the gene was sequenced.
    Response: Thanks for the suggestion, we completely agree with the respected reviewer and clarified thoroughly by adding “exon 2” (line 45, 97, 220, 221, 378 and 503).

  • Line 468: change “thesis,” to “finding”.
    Response: Thanks for this comment. We have changed the wording according to the reviewer’s suggestion (line 509).

  • Line 484: the numbers used in this study are too small to be able to conclude that small and large intestinal tumors “seem to be different on a genomic level with respect to CTNNB1 mutations” (since only 1 small intestinal tumor was sequenced). This statement should be removed.
    Response: Thanks a lot for this reasonable remark. We have removed this statement from the discussion part (line 524-525).

  • Line 497: “Dysregulated β-catenin due to CTNNB1 mutations seem to play a role in feline tumorigenesis, comparable with human and canine intestinal tumorigenesis”. This sentence needs to be reworded as “seem to play a role” is not scientific terminology. Perhaps something like “CTNNB1 mutations are found in feline intestinal carcinomas, as has been reported in human and canine intestinal tumorigenesis”.
    Response: Thanks again for the careful read and valuable advice. We have reworded this conclusive sentence accordingly:
    “CTNNB1 mutations are present in feline intestinal carcinomas, as has been reported in human and canine intestinal tumorigenesis” (line 544-545).